# Kidney Biopsy in a Patient with Cardiorenal Metabolic Syndrome—How to Interpret Histopathology

**Elena Zakharova** [1,2,*] and **Olga Vorobyeva** [3]

1    Department of Nephrology, Botkin Hospital, 125284 Moscow, Russia
2    Department of Nephrology and Hemodialysis, Russian Medical Academy of Continuing Professional Education, 125993 Moscow, Russia
3    Difficult Diagnosis, National Centre of Clinical Morphological Diagnostic, 192071 Saint Petersburg, Russia
*    Correspondence: helena.zakharova@gmail.com; Tel.: +7-967-134-6936

**Abstract:** The components of Cardiorenal Metabolic Syndrome (CRMS) include central obesity, insulin resistance, hypertension, metabolic dyslipidemia, proteinuria, and/or reduced glomerular filtration rate. Kidney biopsy is rarely performed in patients with CRMS; histopathology findings include glomerulopathy, podocytopathy, mesangial expansion and proliferation, glomerular basement thickening, global and segmental sclerosis, interstitial fibrosis and tubular atrophy, and arterial sclerosis and hyalinosis. We report a case of CRMS with slow progression during 10 years of follow-up on chronic kidney disease (CKD). The middle-aged patient had central obesity, hypertension, dyslipidemia, cardiovascular disease, type 2 diabetes mellitus, proteinuria, and CKD stage G3b-G4. Kidney biopsy, performed 3 years after the first presentation, led to the diagnosis of chronic thrombotic microangiopathy (TMA) and complement-associated glomerulopathy. This was not compatible with the medical history and the course of the disease, and previous kidney biopsy review showed metabolic nephropathy with glomerulomegaly, global and segmental glomerulosclerosis, tubular atrophy and interstitial fibrosis, arteriosclerosis, and lipid embolus in the lumen of one artery, and found neither TMA features nor C3 deposition. The reported case demonstrates the importance of an accurate and thoughtful reading and interpretation of kidney biopsy, and stresses that disregarding medical history may potentially mislead and alter the understanding of the true cause of CKD.

**Keywords:** obesity; arterial hypertension; dyslipidemia; diabetes mellitus; cardiovascular disease; chronic kidney disease; kidney biopsy

## 1. Background

The components of Cardiorenal Metabolic Syndrome (CRMS) include central obesity, insulin resistance, hypertension, metabolic dyslipidemia, proteinuria, and/or reduced glomerular filtration rate [1]. Most of the recent publications on CRMS address the issues of pathophysiology, epidemiology, prevention, and care models [2–5]. Much less can be found about kidney pathology pattern(s) and characteristics of CRMS; histopathological findings in metabolic syndrome include a variety of non-specific changes: glomerulopathy, podocytopathy, mesangial expansion and proliferation, glomerular basement thickening, global and segmental sclerosis, interstitial fibrosis and tubular atrophy, and arterial sclerosis and hyalinosis [6–8]. Kidney biopsy is rarely performed in patients with CRMS, and some of the above-mentioned findings are based on autopsy studies [9]. Specifically, obesity-related glomerulopathy refers mainly to podocyte injury with or without focal segmental glomerulosclerosis [10,11], and it has been demonstrated that electron microscopy can reveal lipid droplets in the mesangial and tubular epithelial cells [12].

We report a case of CRMS with metabolic glomerulopathy, diagnosed after the revision of previous kidney biopsy 8 years before, and slowly progressive chronic kidney disease (CKD).

## 2. The Case

A 43-year-old Caucasian male arrived at the tertiary nephrology center in December 2021, complaining of leg pain after 300 m walk, fatigue, and headache, and seeking a second opinion.

His previous medical history included overweight since adolescence and arterial hypertension (AH) since his early 30s, without any work-up for AH origin. At the age of 33 years old, he was admitted to the local internal medicine unit, complaining of dizziness and palpitations; his blood pressure (BP) was 280/120 mm Hg, he had central obesity with BMI 33 kg/m$^2$, and work-up found mild proteinuria (PU) 0.6 g/24 h and elevated serum creatinine (SCr) 213 μmol/L (eGFR 34 mL/min/1.73 m$^2$). The patient received angiotensin-converting enzyme inhibitors (iACE) and calcium-channel blockers (CCB), but four months later his BP was not well controlled, and PU increased to 1.5 g/24 h without worsening of his kidney function. He was diagnosed with CKD stage G3b and referred to the outpatient nephrology unit; work-up included serology tests, kidneys, abdomen and neck ultrasound, and Doppler ultrasound, and ruled out vasculitides, lupus, and renal artery stenosis, but showed hepatosplenomegaly and non-stenotic atherosclerosis of brachiocephalic trunk. He received thiazide diuretics in addition to iACE and CCB.

Two years later, at admission to the secondary nephrology unit, he had moderate PU with blunt urine sediment: white blood cells (WBC) 1–2 hpf, red blood cells (RBC) 0–1 hpf, dysmorphic RBC 0 hpf, hyaline casts 0–1 hpf, and RBC casts 0 hpf. His hemoglobin (Hb) was 173 g/L with mild erythrocytosis (RBC cells 5.8 × 10$^{12}$/L), his SCr remained stable, and he had hyperuricemia, severe dyslipidemia (Table 1) with very-low-density lipoproteins up to 1.59 mmol/L (normal range 0.6–1.1), and elevated C-reactive protein 17.4 mg/L (normal range <8.2). His WBC and platelet count, fasting glucose, serum total protein (STP), serum albumin (SA), serum and urine electrophoresis, liver function and coagulation tests, lactate dehydrogenase (LDH), electrolytes, thyroid-stimulating and parathyroid hormones, immunoglobulins, C3, C4, anti-nuclear, anti-DNA, and anti-neutrophil cytoplasmic antibodies were within normal range, and infectious screening was negative. Kidney ultrasound revealed normal kidney size of 116 × 60 and 119 × 50 mm, and small solitary cysts 6–7 mm in both kidneys. Echo-cardiography (ECHO-CG) found left ventricular hypertrophy with preserved ejection fraction (67%) and ruled out aortic coarctation. Fundoscopy showed hypertensive retinal angiopathy. He underwent kidney biopsy, and according to the pathology report light microscopy found mild mesangial proliferation and widening, mild segmental glomerular basement thickening, global and segmental glomerulosclerosis, moderate focal interstitial fibrosis and tubular atrophy, arterio- and arteriolosclerosis, and mild thrombotic changes in the small arteries. Immunofluorescence showed mild C3 and fibrinogen deposits in mesangium and along capillary walls. Pathology findings were considered as chronic thrombotic microangiopathy (TMA); his anti-β2-glycoprotein antibodies were 20.3 RU/mL (normal range < 20); however, lupus anticoagulant test was negative, ADAMTS13 activity was 70%, haptoglobin was 2.72 g/L (normal range 0.3–2.1), and blood film did not show the presence of schistocytes. The patient was diagnosed with complement-associated glomerulopathy, and advised plasma exchanges.

Three months later, after the first course of plasma exchanges in the local nephrology unit, he developed superficial femoral artery thrombosis and underwent thrombectomy; his laboratory data did not change: PU was 0.7 g/24 h, SCr 185 μmol/L, and serum uric acid 620 μmol/L. Antiplatelet and lipid-lowering agents were added in addition to iACE, CCB, and diuretics.

The following year, after the second course of plasma exchanges, his SCr and PU were stable; however, his BP did not reach the target level. He continued to take iACE, CCB, diuretics, antiplatelets, and lipid-lowering agents.

**Table 1.** Patient's main laboratory data during the follow-up period.

|  | 10.2014 | 11.2019 | 01.2020 | 11.2021 | 02.2022 |
|---|---|---|---|---|---|
| SCr μmol/L (53–115) | 203 | 198 | 225 | 255 | 254 |
| eGFR CKD-EPI mL/min/1.73 m$^2$ | 35 | 35 | 30 | 25 | 25 |
| UA μmol/L (155–428) | 527 | 604 | 621 | 690 | 661 |
| TCh mmol/L (1.3–5.3) | 6.03 | 6.33 | 6.01 | 7.90 | 7.24 |
| HDL mmol/L (1.0–1.55) | 0.76 |  | 0.94 | 1.09 |  |
| LDL mmol/L (1.9–3.3) | 3.74 |  | 3.93 | 5.09 |  |
| TG mmol/L (0.6–2.3) | 3.46 |  | 4.97 | 6.47 |  |
| AC (2.28–3.02) | 7.0 |  | 5.4 | 6.3 |  |
| FG mmol/L (3.5–5.56) | 5.3 | 8.2 | 6.0 | 6.8 | 7.1 |
| PU g/24 h (0.0–0.15) | 0.8 | 2.7 | 4.0 | 0.9 | 0.4 |

Legend: SCr, serum creatinine; eGFR, estimated glomerular filtration rate; CKD-EPI, Chronic Kidney Disease Epidemiology creatinine equation; UA, uric acid; TCh, total cholesterol; HDL, high-density lipoproteins; LDL, low-density lipoproteins; TG, triglycerides; AC, atherogenic coefficient; FG, fasting glucose; PU, proteinuria.

Three and a half years later, he developed Q-wave myocardial infarction, treated without percutaneous coronary intervention; his BP was 200/110 mm Hg, BMI was 38.6 kg/m$^2$, SCr did not change, his PU was moderate, urine sediment was unremarkable, his fasting glucose was elevated (Table 1), and Doppler ultrasound revealed superficial femoral artery occlusion. He was diagnosed with cardiovascular atherosclerotic disease and type 2 diabetes mellitus (DM) and advised to maintain a low-carbohydrate and low-sodium diet; additional treatment included β-blockers, uric acid lowering agents, and dual antiplatelet therapy.

Within the next three months, at admission to the local nephrology unit, his SCr slightly increased, but CKD stage remained at G3b; his PU elevated to nephrotic range, and hyperuricemia, dyslipidemia, and hyperglycemia persisted (Table 1). His total blood count, STP, SA, LDH, and other blood chemistry tests, coagulation tests, immunoglobulins, and complement remained within the normal range. Doppler ultrasound revealed obliterating atherosclerosis of his lower extremity arteries, and ECHO-CG showed preserved ejection fraction (50%). His treatment still included iACE, CCB, β-blockers, diuretics, lipid- and uric-acid-lowering agents, and dual antiplatelet therapy. He was advised to maintain a healthy diet and regular physical activity but did not keep to these recommendations.

Ten months later, he still had severe dyslipidemia, hyperuricemia, and hyperglycemia, and his CKD stage became G4; however, his PU was moderate (Table 1), his STP, SA, and LDH remained normal, and his Hb was 173 g/L.

At referral, he had abdominal obesity, face and neck hyperemia, and decreased pedal pulse; his BP was 150/95 mm Hg; physical examination and vital signs were otherwise normal. He was advised to maintain a healthy diet, his anti-phospholipid antibodies and lipid panel were checked, and his kidney biopsy was sent for review to the tertiary pathology center.

Three months later, his uric acid, total cholesterol, and fasting glucose remained elevated, kidney function was stable (Table 1), and anti-phospholipid antibodies were all negative.

A review of the previous kidney biopsy found 37 glomeruli, 8 of them globally sclerosed. The remaining glomeruli were markedly enlarged, two of them with segmental sclerosis (Figure 1), with single-contoured and non-thickened capillary walls, and without mesangial or endocapillary hypercellularity and crescents. There was moderate interstitial fibrosis and tubular atrophy (20%) and marked tubulointerstitial inflammation. Arteriole walls were thickened, with mild hypertrophy of muscular layer and focal insudative changes; interlobular arterial walls were thickened, with severe intimal fibrosis; and a rounded-oval lipid embolus was found in the lumen of one artery (Figure 2). Immunofluorescence with FITS-conjugated anti- IgA, IgG, IgM, C1q, C3, fibrinogen, and λ and κ antibodies was negative.

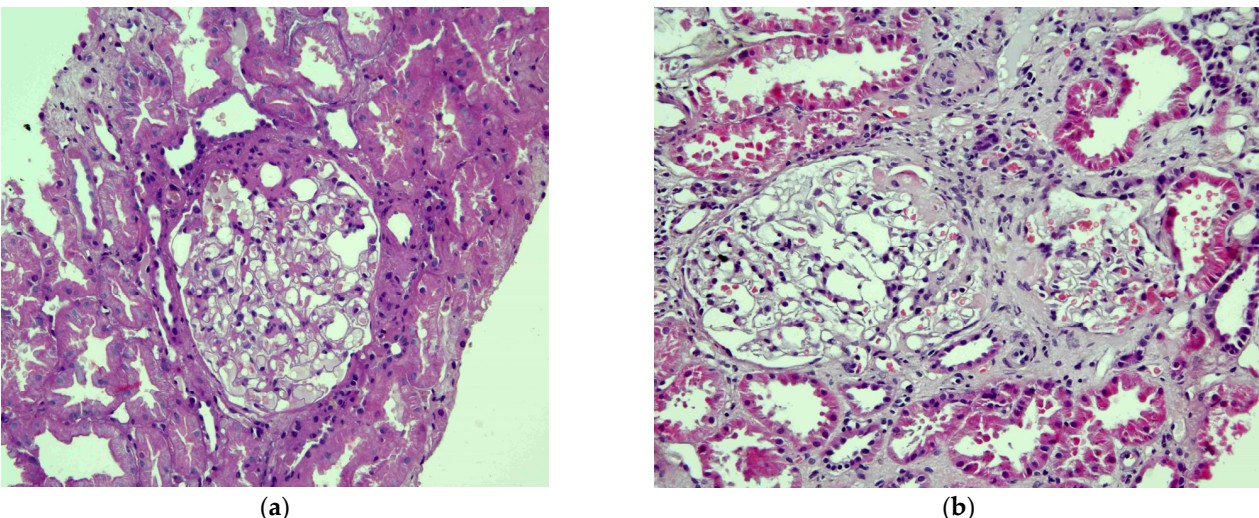

(**a**) (**b**)

**Figure 1.** (**a**) Light microscopy. Marked glomerulomegaly. Hematoxylin & Eosin; magnification ×200. (**b**) Light microscopy. Marked glomerulomegaly with secondary perihilar segmental glomerulosclerosis. Hematoxylin & Eosin; magnification ×200.

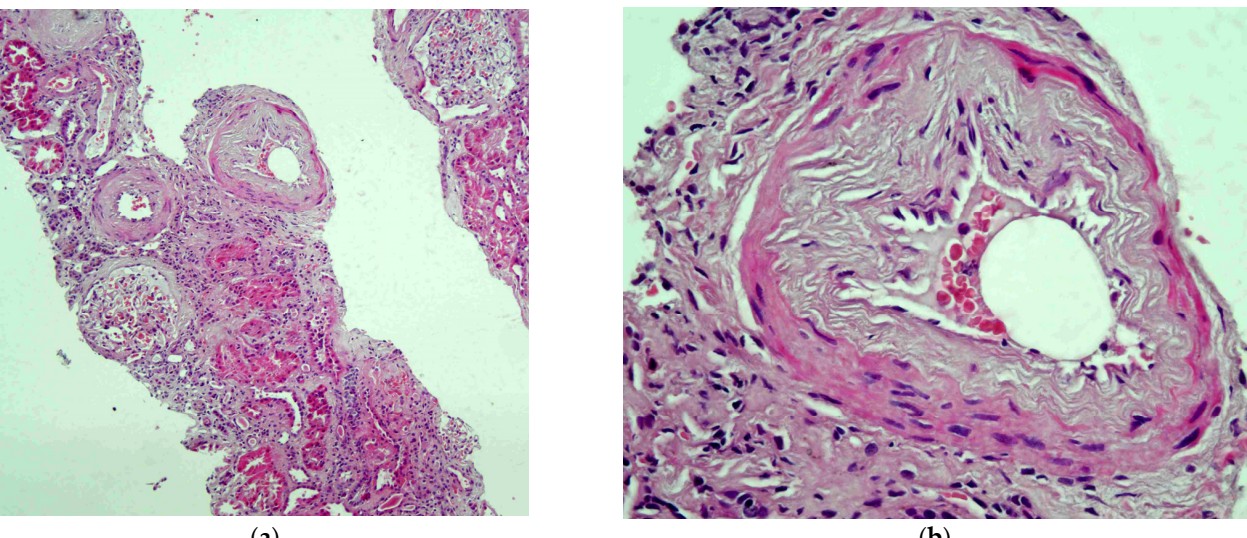

(**a**) (**b**)

**Figure 2.** (**a**) Light microscopy. Glomerulomegaly, tubular atrophy and interstitial fibrosis, and severe arteriosclerosis. An artery with a rounded-oval, clearly delineated, and optically blank inclusion in the lumen (lipid embolus). Hematoxylin & Eosin; magnification ×100. (**b**) Light microscopy. An artery with well-visualized endothelial lining and marked thickening of the wall due to intimal fibrosis. A rounded-oval optically blank inclusion in the lumen, not lined with the endothelium (lipid embolus). Hematoxylin & Eosin; magnification ×200.

Based on the medical history, review of the laboratory data, and revision of kidney biopsy, we diagnosed CRMS with metabolic nephropathy, and advised life-style modification, a switch from iACE to angiotensin receptor blockers (ARBs), and additional treatment with sodium–glucose cotransporter 2 inhibitors, cholesterol absorption inhibitors, and icosapent ethyl.

## 3. Discussion

A severely obese middle-aged male with a 10-year history of CKD, previously diagnosed with complement-associated glomerulopathy, arrived for a second opinion, complaining of intermittent claudication and headache. We retrieved available medical records and concluded that from a clinical perspective our patient met almost all CRMS criteria [1] at the first presentation: he had central obesity, AH, dyslipidemia, proteinuria, and reduced kidney function. He already had non-stenotic atherosclerosis of brachiocephalic trunk, and he might have insulin resistance (despite a glucose tolerance test not being performed) because 7 years later he had overt type 2 DM. Extensive work-up in search of the causes of CKD was attempted twice, and ruled out all suspected associated diseases other than CRMS.

However, CRMS was neglected, and kidney biopsy, performed almost 3 years after the first presentation, found mild mesangial proliferation and widening, mild segmental glomerular basement thickening, global and segmental glomerulosclerosis, moderate focal interstitial fibrosis and tubular atrophy, and arterio- and arteriolosclerosis. All these findings were compatible with those described in metabolic syndrome [6–8]; however, the pathologist reported "mild thrombotic changes" in the small arteries, interpreted as "chronic thrombotic microangiopathy". Even though the patient's Hb and haptoglobin were higher than the upper normal limit, schistocytes were not found, and his platelets and LDH were normal, meaning that he did not meet the TMA diagnostic criteria, which include microangiopathic hemolysis and thrombocytopenia [13,14]; additional work-up was undertaken and found nothing confirmative of TMA. We only can retrospectively assume that the previous diagnosis of "complement-associated glomerulopathy" was based on the mild C3 deposits in the mesangium and along the capillary walls, as reported by the pathologist; and that normal serum C3 probably was not considered as a counter-argument because serum C3 levels may remain normal in some patients with C3 glomerulopathy [15]. We also can speculate that the recommendation to treat the patient with plasma exchanges was based on the literature data, indicating the benefits of plasma exchange in some patients [16,17] and the avoidance of glucocorticoids in a patient with obesity and poorly controlled arterial hypertension. Regardless, two courses of plasma exchanges (the number of procedures and exchanged volumes are not known) did not influence the course of the patient's disease.

Medical treatment did not control the patient's BP nor dyslipidemia, not least because of his sedentary life-style and non-adherence to the recommended diet. Eight years after the kidney biopsy, the patient had persistent severe metabolic dyslipidemia with extremely high atherogenic coefficient and triglyceride levels, and developed atherosclerotic polyvascular disease with Q-wave myocardial infarction and obliterating atherosclerosis of his low extremity arteries. Of note, he also had persistent hyperuricemia, which is an important part of CRMS pathogenesis [1]. Taken together, central obesity, type 2 DM, hypertension, metabolic dyslipidemia, proteinuria, reduced kidney function, hyperuricemia, and atherosclerotic cardiovascular disease were definitely compatible with CRMS diagnosis. However, given that the patient demonstrated arterial hypertension since the age of 30, myocardial infarction at the age of 41, and anti-β2-glycoprotein antibodies slightly exceeding the upper normal limit, we checked the full spectrum of the anti-phospholipid antibodies, which were all within the normal range, and ruled out anti-phospholipid syndrome. On the other hand, the patient presented with mild-to-moderate proteinuria with blunt urine sediment, and SCr 230 μmol/L with a very slow rate of CKD progression. At presentation, he had CKD stage G3b, and progressed towards CKD stage G4 with estimated GFR decline

from 34 to 25 mL/min/1.73 m$^2$ over 10 years. This is not typical for C3 glomerulopathy, characterized not only by proteinuria, but also by micro- or macroscopic hematuria and progressive reduction of kidney function [17].

All these considerations demanded revision of the complement-associated glomerulopathy diagnosis, and we decided to review the previous kidney biopsy. We found glomerulomegaly, global and segmental glomerulosclerosis (27%), moderate tubular atrophy and interstitial fibrosis, severe arteriosclerosis, and lipid embolus in the lumen of one artery, consistent with the diagnosis of metabolic nephropathy with secondary global and focal glomerulosclerosis. Of note, despite electron microscopy not being performed, we were able to identify lipid embolus in the artery lumen using light microscopy. We did not find any thrombi, and immunofluorescence was completely negative. Again, we can only speculate that focal insudative changes in the arterioles were misinterpreted as thrombotic changes, and that mild non-specific C3 and fibrinogen fluorescence were considered as true C3 and fibrinogen deposition.

Our treatment recommendation, according to the Multispecialty Practice Recommendations for the management of diabetes, cardiorenal, and metabolic diseases [5], included sodium–glucose cotransporter 2 inhibitors, cholesterol absorption inhibitors, and icosapent ethyl in addition to ARBs, CCB, diuretics, and dual antiplatelet therapy.

## 4. Conclusions

The reported case demonstrates the importance of an accurate handling and reading of kidney biopsy, and stresses that the formal clinical interpretation of pathology findings disregarding medical history, physical examination, and laboratory findings is potentially misleading and may alter the understanding of the true cause of CKD, and, as a consequence, lead to unnecessary aggressive treatment.

**Author Contributions:** Conceptualization, E.Z.; writing—original draft preparation, E.Z.; writing—review and editing, E.Z. and O.V.; visualization, O.V. All authors have read and agreed to the published version of the manuscript.

**Funding:** This research received no external funding.

**Institutional Review Board Statement:** Ethical review and approval were waived for this study due to its retrospective non-interventional nature.

**Informed Consent Statement:** Written informed consent was obtained from the patient to publish this paper.

**Data Availability Statement:** Data presented in this study are available on the request from the corresponding author. The data are not publicly available due to the privacy reasons.

**Conflicts of Interest:** The authors declare no conflict of interest.

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
