# Peer review of "Kidney Biopsy in a Patient with Cardiorenal Metabolic Syndrome—How to Interpret Histopathology"

_kidneydial, doi:10.3390/kidneydial3020015_

Round 1

Reviewer 1 Report

Thank you for submission.

Zakharova and Vorobyeva report an interesting case of a 43 year old with a cardiorenal metabolic syndrome who had a prior diagnosis of complement-associated glomerulopathy and chronic TMA based on renal biopsy. Re-analysis of results have indicated the kidney disease is more likely due to a metabolic syndrome.

Well done to the authors for providing a thorough timeline of events from the patient's first presentation and for re-analysing results of the kidney biopsy previously completed.

Comments:

1. For the urine results, you have reported and tracked the patient's proteinuria with thanks. Did the patient also have any haematuria (if so, were the any dysmorphic red cells)? And was there any analysis for urine casts such as red cell casts?

2. At the time when the patient was diagnosed with a chronic TMA, was the patient's blood film examined for any schistocytes/red cell fragments?

3. For the light microscopy, are you able to include a scale bar on the images, if possible?

4. Was there any renal imaging completed (ultrasound or CT)? What size were the kidneys and any other relevant findings on the imaging?

5. The was diagnosed at hypertension at quite a young age (30s). Did he previously have an work up for secondary hypertension?

Author Response

  1. For the urine results, you have reported and tracked the patient's proteinuria with thanks. Did the patient also have any haematuria (if so, were the any dysmorphic red cells)? And was there any analysis for urine casts such as red cell casts?

Thank You very much for this comment. We have indicated in the text, that urine sediment was blunt. We have added the details:  2014 – WBC 1-2 hpf, RBC 0-1 hpf, dysmorphoc RBC 0 hpf, hyaline casts 0-1 hpf, RBC casts – 0 hpf; 2019 WBC 2-3 hpf, RBC 0-2 hpf, no casts; 2021 - WBC 1-2hpf, RBC 0-2hpf, no casts; 2022 WBC 1-2, RBC 0-1, no casts

  1. At the time when the patient was diagnosed with a chronic TMA, was the patient's blood film examined for any schistocytes/red cell fragments?

Thank You very much for Your comment. At the time of TMA diagnosis patient’s blood film was examined, and schisocytes were not found, we’ve added this information

  1. For the light microscopy, are you able to include a scale bar on the images, if possible?

Thank You very much for this comment. Unfortunately, taking the microphotographs we did not use the scale bar

  1. Was there any renal imaging completed (ultrasound or CT)? What size were the kidneys and any other relevant findings on the imaging?

Thank You very much for the comment. Kidney ultrasound was performed, as it is indicated in the text, and was unremarkable, right kidney size was 116 x 60 mm, left kidney size 119 x 50 mm, solitary cysts 6-7 mm in both kidneys, we have added this information.

  1. The was diagnosed at hypertension at quite a young age (30s). Did he previously have an work up for secondary hypertension?

Thank You very much for this comment. The patient did not have any work-up before 2012, later on Doppler ultrasound ruled out renovascular origin of hypertension, as it is indicated in the text; his TSH, sodium and potassium levels were normal, as it is indicated in the text, and ECHO-CG ruled out aortic coarctation (we have added this information).

Reviewer 2 Report

Interesting case report study. Minor spell check, considering English are required. In line 58 and 100, authors are writing about dyslipidemia present in this patient, but, considering the Table 1, data for this time period is missing (08/2012, 01.2015.) Authors should fill in the table if they have this data, or avoid to mention dyslipidemia in the manuscript if they do not have. With a lot of missing data, table 1 does not look nice, neither scientific, nor techical.  Unmark references 6,7,9.

Author Response

Interesting case report study. Minor spell check, considering English are required. In line 58 and 100, authors are writing about dyslipidemia present in this patient, but, considering the Table 1, data for this time period is missing (08/2012, 01.2015.) Authors should fill in the table if they have this data, or avoid to mention dyslipidemia in the manuscript if they do not have. With a lot of missing data, table 1 does not look nice, neither scientific, nor techical.  Unmark references 6,7,9.

Thank You very much for Your comments. We do not have lipids level numbers for 2012 and 2015, therefore, we excluded mentioning of dyslipidemia in the relevant paragraphs, We also re-worked the table and deleted columns and rows with missing data. As per Your advice we checked spelling and unmarked references 6, 7 and 9

Reviewer 3 Report

In this report, the authors aimed to emphasise the importance of an accurate and thoughtful reading and interpretation of the renal biopsy for differential diagnosis in the course of cardio-renal metabolic syndrome. The report is accurate in its presentation but does not add scientific value to the field of renal biopsy.

Author Response

In this report, the authors aimed to emphasise the importance of an accurate and thoughtful reading and interpretation of the renal biopsy for differential diagnosis in the course of cardio-renal metabolic syndrome. The report is accurate in its presentation but does not add scientific value to the field of renal biopsy.

Thank You very much for Your comment. In fact, considering the topic of the Special issue, we did not intend to add scientific value to the field of kidney biopsy; we just aimed to stress the importance of an accurate handling and reading of kidney biopsy, and its clinical interpretation considering physical examination, medical history and laboratory data. We have added this point to the Conclusions.

Round 2

Reviewer 1 Report

Thank you for addressing the comments.

Reviewer 2 Report

The authors have successfully responded to all given comments and objections.

Reviewer 3 Report

The authors improved the quality of the manuscript, It can be suitable for publication.

Round 3

Reviewer 2 Report

The authors have successfully responded to all given comments and objections.